# The Never Ending Story—What Are the Differentiable Magnetic Resonance Imaging Characteristics Between Pyogenic and Mycobacterial Thoracolumbar Infections?

**DOI:** 10.3390/jcm14020318

**Published:** 2025-01-07

**Authors:** Marcin Waśko, Jerzy Białecki, Oleg Nowak, Agnieszka Kwiatkowska-Miernik, Agata Bujko-Małkiewicz, Jerzy Walecki

**Affiliations:** 1Department of Radiology and Imaging, The Medical Centre for Postgraduate Education, 01-813 Warsaw, Poland; marcin@wasko.md; 2Independent Public Clinical Hospital Named After Prof. Adam Gruca of the Center for Postgraduate Medical Education, 05-400 Otwock, Poland; kl.ortopedii@spskgruca.pl; 3Department of Diagnostic Imaging, SPSK Im. A. Grucy, 05-400 Otwock, Poland; 4Department of Radiology, Radiotherapy and Nuclear Medicine, National Medical Institute of the Ministry of the Interior and Administration, 02-507 Warsaw, Poland; akwiatkowskamiernik@gmail.com (A.K.-M.); agata.bujko-malkiewicz@hotmail.com (A.B.-M.)

**Keywords:** spondylodiscitis, mycobacterial, pyogenic, infection, bacterial, magnetic resonance imaging

## Abstract

**Background/Objectives**: This study aimed to determine if MRI features can distinguish between spinal infections caused by pyogenic bacteria and Mycobacterium tuberculosis. **Methods**: Patients underwent an MRI of the thoracolumbar spine with and without contrast. Three blinded observers assessed the images, using statistical tests for analysis. **Results**: Demographic characteristics and symptom duration were similar between patients with tuberculous and pyogenic spinal infections. In 36 cases of pyogenic infections, the MRI showed weakly delineated paravertebral tissue enhancement (76%), a hyperintense signal in the T2 TIRM sequences (89%), and homogeneous vertebral body enhancement (89%). In 32 cases of Mycobacterium infections, the MRI revealed well-delineated paravertebral changes, mixed vertebral body signals, and variable enhancement. Pyogenic infections were more often found in the lumbar spine (67%) and typically involved two vertebrae, while tuberculous infections preferred the thoracic spine (75%) and often involved two vertebrae, with 25% affecting three or more vertebrae. **Conclusions**: The MRI features can help differentiate between pyogenic and tuberculous spine infections, though none are definitive. The study suggests that MRI can be used for initial differentiation or as a diagnostic tool when biopsy or surgical exploration is not possible.

## 1. Introduction

Differential diagnosis of inflammatory vertebral lesions remains a challenge for orthopedic surgeons and radiologists [1]. The ambiguous clinical course often modified by previous therapies leads clinicians to refer their patients for imaging. A classic radiographic examination reveals the narrowing of the intervertebral spaces and irregular outlines of the border plates, but the lesions are not visible until a few weeks after the onset of symptoms [2]. Therefore, radiography carries low specificity and relatively low sensitivity, especially in the early stages of spinal infections [3,4]. Computed tomography can visualize sclerosis of the vertebral bodies, bone block formation, destruction of the vertebral bodies, and the presence of soft-tissue edema and abscesses, and has slightly higher sensitivity than a standard radiograph [4]. The efficacy of radiography and computed tomography in the early diagnosis and differential diagnosis of spinal infections is, therefore, minimal [5,6].

Magnetic resonance imaging (MRI) offers significant advantages in diagnosing spinal infections due to its superior soft-tissue contrast and multiplanar capabilities. It enables early detection of infections by visualizing bone marrow edema, soft-tissue involvement, and abscess formation before these changes are apparent in other imaging modalities [7]. Additionally, MRI can help differentiate between pyogenic and tuberculous spondylodiscitis by identifying characteristic patterns such as paraspinal abscesses or vertebral body signal changes, thus facilitating more accurate and timely diagnoses [8,9].

Bone scintigraphy serves as a cost-effective and highly sensitive diagnostic tool for localizing lesions in suspected cases of tuberculous spondylodiscitis [10]. Its ability to detect clinically silent lesions makes it a valuable asset for the treating surgeon. Additionally, the presence of associated typical rib lesions and the absence of skull involvement can provide supplementary clues in ambiguous cases, aiding in the diagnostic process. However, its limitations, including non-specificity and the potential for false-negative results, restrict its utility to that of a supportive rather than definitive diagnostic method in cases of suspected tuberculous spondylodiscitis [11,12].

Before laboratory confirmation and final diagnosis, the inflammatory process can progress significantly, making treatment difficult and prognosis unfavourable. Therefore, early diagnosis allows for the implementation of treatment and protects the patient from irreversible consequences. These may include neurological disorders, vertebral deformity, and the resulting disability [13]. There is also a subset of patients who cannot be treated surgically because of contraindications to anesthesia [14]. Negative biopsy results have also been published, despite the presence of disease [15].

The standardized treatment of spondylodiscitis is strongly recommended to ensure patients achieve and maintain a good quality of life. Among the various treatment schemes proposed, the most recent by Pola et al. stands out as a comprehensive approach [16]. This scheme incorporates all available orthopedic interventions, enabling spine surgeons to minimize complications, reduce costs, and avoid overtreatment, thus offering significant benefits for both patients and healthcare systems. However, clinicians and radiologists still face considerable challenges in the accurate differentiation between pyogenic and tuberculous spondylodiscitis, particularly in the early stages. This distinction is critical, as the two forms require different therapeutic approaches, yet overlapping clinical and radiological features often complicate diagnosis. As a result, delays in appropriate treatment remain a significant obstacle, emphasizing the need for further research and diagnostic tools to improve early and precise differentiation [17].

The aim of this study was to evaluate the utility of MRI, in differentiating between pyogenic and tuberculous spondylodiscitis. We hypothesized that MRI may provide distinct signal characteristics that can reliably differentiate these two types of spinal infections, enabling earlier and more accurate diagnosis.

## 2. Materials and Methods

### 2.1. Patients

The study cohort consisted of 68 consecutive patients with suspected thoracic-lumbar spine infections treated between January 2015 and December 2019 at the SPSK im. prof. A. Grucy in Otwock. Patients with a history of spine surgeries were excluded from the study. Among 78 patients hospitalized with a preliminary diagnosis of thoracic-lumbar spine infection in the study period, 9 were diagnosed with tumour metastases, and full radiological documentation was not available for 1 patient. An institutional review board approved this study before initiation.

### 2.2. Imaging Examinations

All patients underwent a standard X-ray of the thoracolumbar spine in two perpendicular projections: anteroposterior and lateral. All patients underwent MRI of the thoracolumbar spine with and without contrast with a 1.5-T scanner (Siemens Essenza, Erlangen, Germany) before starting the therapy. T1 (repetition time/echo time [TR/TE], 691–729/10–11), T1 FS (TR/TE, 505–550/10–11), and T2 turbo inversion recovery magnitude (TIRM) (TR/TE, 5350/86) images, as well as T1-weighted sequences after contrast administration, were obtained in three orthogonal planes (sagittal, frontal, and axial). Details are provided in Appendix A.

Three independent observers—two radiologists and one orthopedic surgeon with 10 years of experience—assessed the images. All readers were blinded to the diagnosis. Quantitative variables were averaged between measurements. Qualitative variables were noted. Discrepancies were resolved through consensus.

The number and location of the involved segments were evaluated. The vertebral bodies’ signal and intervertebral discs were assessed. Signal changes in T1- and T2-weighted images were described by comparing the diseased segment (segments) with healthy tissue, defining the signal as hypointense, isointense, hyperintense, mixed, or fluid (the last option only for T2-weighted images). In the case of vertebral bodies, the extent of signal change was determined in quartiles (1–25%, 26–50%, 51–75%, and 76–100%) [18].

After contrast administration, signal enhancement of vertebral bodies and intervertebral discs in T1-weighted images was evaluated (even, uneven, marginal, or none). The extent of enhancement in the vertebral bodies was determined in quartiles. The delimitation of signal enhancement in paravertebral soft tissues (strong or weak) was assessed. The enhancement within the paravertebral soft tissues was also evaluated. The presence of soft-tissue abscesses was assessed separately. We reported meningeal enhancement at the lesion level and the presence or absence of epidural abscess.

The last group of parameters evaluated comprised the indicators of morphological destruction. The percentage loss of vertebral body height (categorizing results into quartiles), the status of endplates (unchanged, erosions, or destruction), and the destruction of the intervertebral disc were assessed. The destruction of the intervertebral disc was determined on the following scale: none; initial (increase attributable to the expansion of the intervertebral disc with a change in signal); moderate (reduction to 50% of disc height compared with the healthy segments); significant (reduction above 50%); and total or fluid signal within disc on T2-weighted images [18].

### 2.3. Hospital Stay and Data Collection

The diagnosis of spinal infections was made according to the Infectious Disease Society of America guidelines (Clinical Practice Guidelines for the Diagnosis and Treatment of Native Vertebral Osteomyelitis in Adults) [19]. The following data were collected: age, body mass index, gender, initial and final clinical diagnosis, and results of imaging tests. This study was retrospective and non-interventional.

### 2.4. Statistical Analysis

Continuous variables were presented as mean with 95% confidence interval, and minimum and maximum or median with the interquartile range. Student’s *t*-test compared the mean values. The normality of distribution was tested with a Shapiro–Wilk test. In the absence of normality of the distributions of the studied variables, the mean values were compared by the Mann–Whitney U test. Frequencies were compared by Fisher’s exact test for 2 × 2 tables and Cramér’s V coefficient for larger tables. Statistical analysis was performed using MedCalc 18.5 software (MedCalc, Ostend, Belgium). The threshold for statistical significance was set at 0.05.

## 3. Results

### 3.1. Characteristics of the Study Group

Pyogenic bacteria were detected in 36 patients (24 men and 12 women). The average age in this group was 62 years (95% confidence interval, 58 to 67; range, 38–83). The median period from onset to hospitalization was 4 months (range, 3–12). Spinal tuberculosis was diagnosed in 32 patients (18 men and 14 women). The average age in this group was 51 years (95% confidence interval, 45 to 58; range, 24–79). The median period from onset to hospitalization was 5 months (range, 5–12). Patients with tuberculosis were 10 years younger (95% confidence interval for the difference, 4 to 18; Student’s *t*-test, *p* = 0.004). Median time from onset of symptoms to hospitalization did not differ between groups (*p* = 0.372).

### 3.2. Advanced Imaging Parameters

Unlike patients with tuberculous infections, the lesions in patients with pyogenic bacteria were significantly more often located in the lumbar region (67%) than in the thoracic spine (33%) (*p* = 0.007) (Table 1). All patients suffering from pyogenic bacteria had only two vertebrae involved. Infection involving three vertebrae and more occurred in one-quarter of patients with tuberculosis (*p* = 0.039) (Table 2).

The distribution of vertebral body signal changes on T1-weighted images was not significantly different between groups (*p* = 0.214). However, a difference was observed on T2 TIRM images (*p* = 0.001) (Table 3). The majority of pyogenic bacterial infections had a hyperintense signal (89%), whereas a heterogeneous signal characterized tuberculosis (69%). Changes in the signal of intervertebral discs had a similar distribution on T1-weighted images (*p* = 0.229) and T2 TIRM images (*p* = 0.245). Distributions of the range of signal changes in the vertebral bodies did not differ significantly between T1-weighted and T2 TIRM images (*p* = 0.308). An example of the features of a pyogenic infection is shown in Figure 1.

After contrast administration, the distribution of the vertebral body signal on T1-weighted images changed significantly (*p* = 0.003) (Table 4). Homogeneous signal amplification dominated in pyogenic bacterial infections (89%). In patients with tuberculosis, mixed amplification in 20 patients and homogeneous amplification in 12 were noted. The distribution of intervertebral disc signal enhancement was not significantly different between groups. In most patients, this amounted to marginal strengthening (56%) (*p* = 0.164). The distribution of the strengthening range in the vertebral bodies did not differ significantly between groups (*p* = 0.745). The signal from paravertebral soft tissues was enhanced after contrast administration in the vast majority of patients (97%), and the groups did not differ in this respect (*p* = 1.000). No signal enhancement in soft tissues was observed in any patient with a pyogenic bacterial infection. Tissue enhancement was well demarcated in tuberculosis patients (94%). In most pyogenic bacterial infections (72%), tissue enhancement was poorly demarcated (*p* < 0.001) (Table 5). Soft-tissue abscesses were slightly more common in tuberculosis infections (*p* = 0.045) (Table 6). Meningeal enhancement was significantly more common in tuberculosis infections (75% compared with 28% in pyogenic bacterial infections) (*p* = 0.015). An epidural abscess was significantly more common in tuberculous infections, 56% compared with 11% in pyogenic bacterial infections (*p* = 0.009) (Table 7). An abscess was evident in 18 patients (56%) with tuberculosis and in only 4 patients (11%) with infection caused by pyogenic bacteria.

Tuberculosis was associated with more intense destruction of vertebral bodies. Most patients in this group (75%) had advanced destruction of more than 50% in height, while 56% of patients with pyogenic bacteria had a loss of between 25% and 50% in height (*p* = 0.016). The condition of the border plates did not differ between the groups (*p* = 0.152); all patients had erosions or destruction. There were also no differences in the level of destruction of the intervertebral disc (*p* = 0.560). The fluid signal within the intervertebral disc was observed in 12 patients with pyogenic bacterial infection and in 6 patients with tuberculous infection (*p* = 0.448). An example of the features of a pyogenic infection is shown in Figure 2.

## 4. Discussion

The most important finding of the current study is the observation of significant, though not absolute, differences in MRI findings between patients with infections caused by pyogenic bacteria and those suffering from spinal tuberculosis. The most clearly outlined differences included lumbar involvement (67% vs. 19% for purulent spondylitis and tuberculosis, respectively), the presence of poorly delimited strengthening of the paravertebral tissue (76% vs. 6%), uniform contrast enhancement (89% vs. 38%), less advanced destruction of vertebral bodies, and hyperintense/homogeneous vertebral signal in T2 TIRM images (89% vs. 31%). In contrast, tuberculosis occurred most frequently in the thoracic spine (75% compared with 22% in tuberculous spondylitis), with more severe destruction of vertebral bodies, heterogeneous vertebral body signal on T2 TIRM images (69% vs. 11%), excellent delineated enhancement of the paravertebral tissue contrast (94% vs. 24%), the presence of paravertebral abscess (75% vs. 39%), strengthening of the signal from the spinal meninges at the level of the affected segment of the spine (75% vs. 28%), and epidural abscess formation (56% vs. 11%).

Signal parameters from the vertebral body and intervertebral discs, also after contrast administration, were not useful in differentiating the etiology of infection. The only exception was the signal of vertebral bodies on the T2-weighted TIRM sequences, whereby the majority of patients with pyogenic bacteria had a hyperintense signal (89%) and in tuberculosis a mixed signal (69%), but this was also not an absolute difference. No significant differences were found in the pattern of contrast enhancement of the intervertebral discs, degree of disc destruction, occurrence of intervertebral disc abscesses, and differences in the degree of destruction of the edge plates.

Differential diagnosis of spinal infection based on imaging tests is not without flaws. A classic X-ray examination shows changes only after a few weeks of disease development and bears low specificity and relatively low sensitivity [2,3,20]. Computed tomography is also not conclusive in the differential diagnosis [4,5,6]. In each case, when a biological test is possible (biopsy from the site), microbiological evaluation is preferred [21,22,23]. However, there is a group of patients with contraindications (e.g., patients with severe, uncontrolled comorbidities) who cannot undergo this procedure [24]. In these patients, radiological diagnostics will be of particular value because both types of inflammation may present similar symptomatology [9,25,26].

The available literature describes features that help to differentiate the etiology of spinal infection using MRI [27,28,29]. Jung et al. showed the presence of well-delimited areas of contrast enhancement in paravertebral soft tissues, more frequent abscesses, involvement of the thoracic segment, and involvement of several levels of tuberculosis in the spine [30]. Chang et al. observed more advanced destruction of the spine in tuberculosis infection and mixed vertebral body signal after administration of contrast in tuberculosis with uniform contrast enhancement from purulent inflammation [18]. de Souza et al. also emphasized the importance of the spinal cord enhancement pattern after administration of contrast agents as a differentiating feature [25]. Foti et al. concluded that disc edema and paraspinal involvement represent the most reliable MRI parameters for diagnosis and differential diagnosis [31]. The development of advanced contrast agents, such as hyaluronic acid-coated radiotracers, offers potential for future applications in spondylodiscitis, further enhancing diagnostic precision and aiding in the differentiation of its etiology [32].

In the present study, we showed that purulent infections of the spine in the lumbar region are more common, as previously described in numerous scientific reports [18,33,34]. One of the features more frequently observed in tuberculous infections was meninges involvement. Similar results were reported previously [18,34,35]. Likewise, Sharif concluded that the spread of pyogenic bacterial infections along the meninges is rare [34].

In our cohort, the involvement of more than two vertebrae was observed only in tuberculosis infection of the spine, as observed by other authors [30,33,36,37]. Most likely this is related to the fact that the tuberculosis process begins in the anterior part of the vertebral bodies, causing their destruction and abscess formation under the anterior longitudinal ligament, which then spreads to adjacent segments [18,35,38].

Many studies published so far highlight that spine tuberculosis spares intervertebral discs. The mycobacteria do not produce proteolytic enzymes [39,40,41]. Thus, the intervertebral disc remains undamaged for a long time in the case of tuberculosis infection [42]. In the current study, intervertebral disc destruction was similar in both types of infection. This finding may be due to the relatively long duration of the disease before MRI, as was the case in the study by Jung et al. [30].

The observed differences in vertebral body signal on T2 TIRM images have not been described thus far. In tuberculous infections a mixed signal was observed, whereas in pyogenic bacterial infection the signal was homogeneously hyperintense. Jung et al. reported a higher incidence of hyperintense vertebral body signal on T2-weighted images in tuberculosis of the spine; however, they did not observe signal heterogeneity [30]. The reason for this phenomenon remains unclear. A possible cause is collapse of bone trabecula during the destruction of the body and sclerosis of the border plates resulting from tuberculosis [33].

This study has several limitations. Despite the relatively large study group, the results reflect the experience of a single centre. Also, the ratio between patients with purulent and tuberculous infections was almost 1:1, which is not typical for this disease [43,44]. This proportion is due to the profile of our hospital, which is the reference centre for musculoskeletal tuberculosis. Another limitation is the relatively long time from onset to hospitalization. The prolonged diagnostic and therapeutic processes have been described in patients with spondylitis, especially in the course of tuberculosis [45]. It is also possible that some of the observed anatomical differences between groups can be explained by patients’ comorbidities and pre-existing changes in, for example, bone marrow signals related to the ageing process, diabetes, and other medical conditions [46,47,48,49]. Nevertheless, it would be even more striking to analyze MRI in the early stages of inflammation, at which point the differences between various etiologies could be more pronounced [50,51]. Last, this study was performed on a 1.5-T scanner, although more powerful scanners are available and might offer more robust imaging of tissues. Moreover, the extent of signal change was determined in subjective quartiles, which is a common approach in medical research. For research purposes, one could use absolute relaxation time constants provided by the imaging workstation, which might allow the use of more robust statistical tests [52]. However, the analysis of absolute relaxation time constants might be user dependent (regarding selection of the region of interest), time-consuming, and not readily available for everyday clinical use. Additionally, the relaxation values from this study could not be extrapolated to newer 3-T MRI settings.

This study provides a comprehensive, though partly subjective, analysis of multiple MRI features found in spondylodiscitis. These features allow for differentiation between pyogenic and tuberculous spondylodiscitis before employing invasive diagnostic methods. Further research employing more robust MRI technologies is warranted to enable the adaptation of research strategies for clinical use (e.g., using relaxation time constants or diffusion-based techniques) [53,54,55,56].

## 5. Conclusions

The comparison of MR images from patients diagnosed with tuberculosis and purulent spondylitis allowed us to identify the features of initial differentiation between these two entities. For tuberculous spondylitis, these include more frequent involvement of the thoracic spine, more severe destruction, well-delimited areas of strengthening soft vertebral tissues, heterogeneous vertebral body signal in tuberculosis, and more frequent occurrence of soft-tissue paravertebral abscesses, epidural abscesses, and meningeal signal enhancement. In spinal infections caused by pyogenic bacteria, lumbar involvement, poor demarcation of the paravertebral tissue-strengthening areas, and a uniform, hyperintense vertebral body signal on T2 TIRM images were more frequently observed. The differences observed in this study might be sufficient for diagnosing the type of infection in patients who, for any reason, are not optimal candidates for surgical biopsy or have inconclusive results from other diagnostic studies.

## Figures and Tables

**Figure 1 jcm-14-00318-f001:**
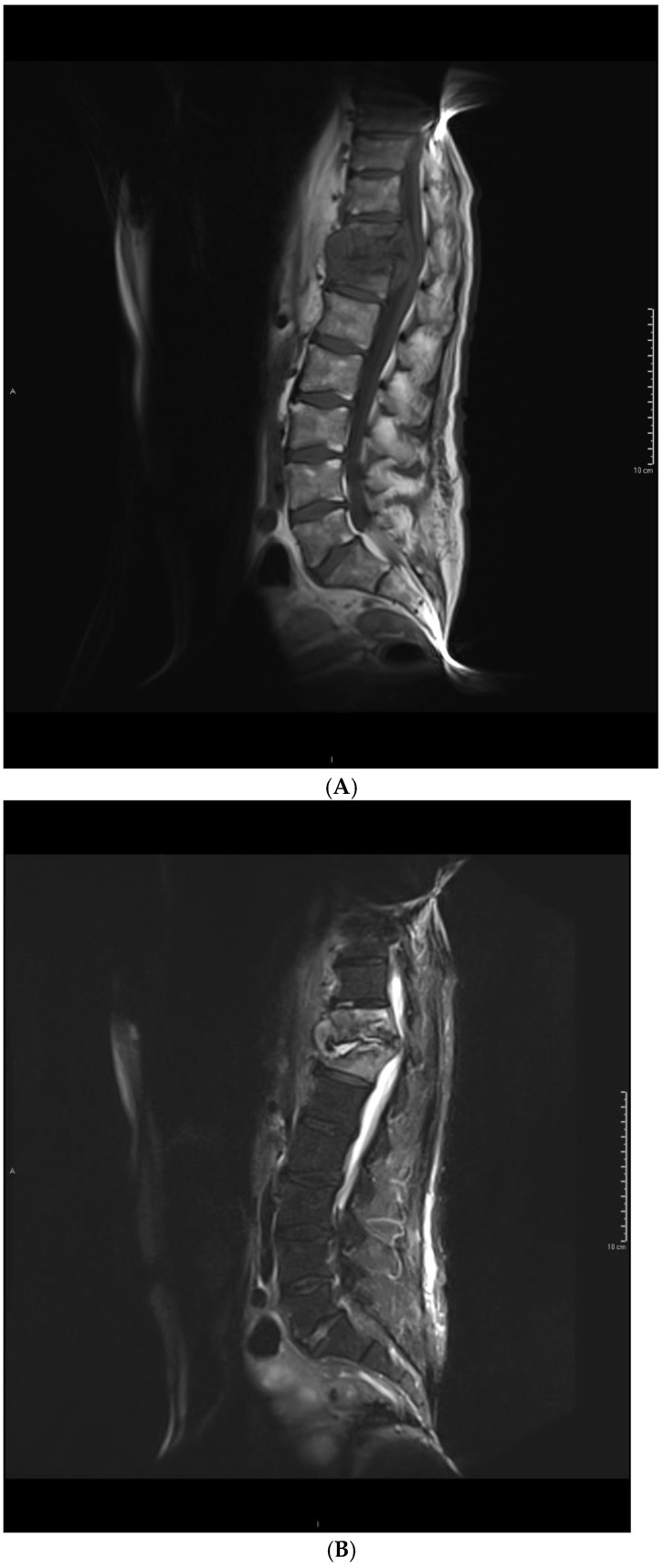
An example of pyogenic spondylodiscitis imaging features in a 68-year-old patient. Images in (**A**) T1 sagittal, (**B**) T2 TIRM sagittal, and (**C**) T1 sagittal with contrast sequences. There is inflammatory involvement of Th11 and Th12 vertebral bodies and an abscess in the Th11–Th12 disc.

**Figure 2 jcm-14-00318-f002:**
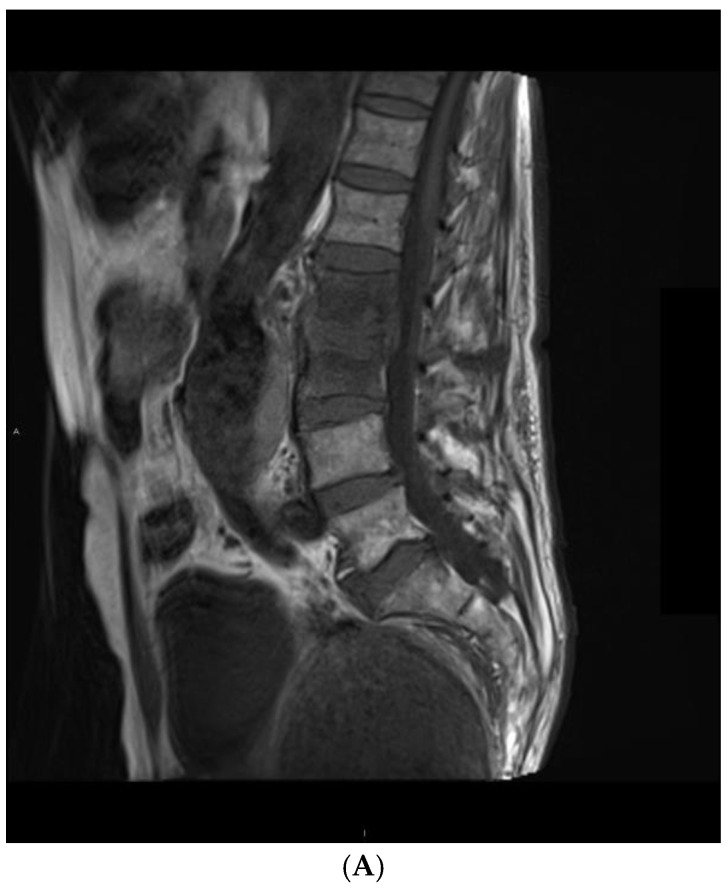
An example of mycobacterial spondylodiscitis imaging features in a 41-year-old patient. Images in (**A**) T1 sagittal with contrast, (**B**) T2 TIRM sagittal, and (**C**) T1 axial with contrast sequences. There is inflammatory involvement of Th3 and Th4 vertebral bodies, paraspinal abscess, epidural abscess, and partial destruction of the vertebral bodies.

**Table 1 jcm-14-00318-t001:** Spine region involved in infection.

Region of Spine	Pyogenic Infection (n)	Mycobacterial Infection (n)
Thoracic	12	24
Lumbar	24	8

**Table 2 jcm-14-00318-t002:** The number of vertebral bodies involved in infection.

Vertebral Bodies Involved	Pyogenic Infection (n)	Mycobacterial Infection (n)
2	36	24
≥3	0	8

**Table 3 jcm-14-00318-t003:** Distribution of vertebral body signal changes in T2 TIRM images.

Type of Signal Distribution	Pyogenic Infection (n)	Mycobacterial Infection (n)
Hyperintense	32	10
Heterogeneous	4	22

**Table 4 jcm-14-00318-t004:** Type of signal enhancement of the vertebral body signal in T1 images after contrast administration.

Type of Signal Enhancement	Pyogenic Infection (n)	Mycobacterial Infection (n)
Homogeneous	32	12
Heterogeneous	4	20

**Table 5 jcm-14-00318-t005:** Type of paravertebral soft-tissue signal enhancement after contrast administration.

Type of Signal Enhancement	Pyogenic Infection (n)	Mycobacterial Infection (n)
None	2	0
Present	34	32

**Table 6 jcm-14-00318-t006:** Occurrence of soft-tissue abscess.

Soft-Tissue Abscess	Pyogenic Infection (n)	Mycobacterial Infection (n)
None	22	8
Present	14	24

**Table 7 jcm-14-00318-t007:** Occurrence of epidural abscess.

Epidural Abscess	Pyogenic Infection (n)	Mycobacterial Infection (n)
None	32	14
Present	4	18

## Data Availability

The data presented in this study are available on request from the corresponding author due to legal reasons (GDPR).

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
