# Peer review of "The Never Ending Story—What Are the Differentiable Magnetic Resonance Imaging Characteristics Between Pyogenic and Mycobacterial Thoracolumbar Infections?"

_jcm, 2025, doi:10.3390/jcm14020318_

Round 1

Reviewer 1 Report

Comments and Suggestions for Authors

Overall this is a well written article which reviews the literature on Pott Disease and pyogenic discitis. 

I think that three aspects should be pointed out by the authors and additional references included regarding: a) classification of pyogenic spondylodiscitis and management altorithm, b) role of nulcear medicine in Pott Disease, c) importance of enhancing contrast agents as a potential for further optimization of diagnostics in neuroradiology and nuclear medicine.

Please see the studies  that you could include to make those points:

Pola E, Autore G, Formica VM, Pambianco V, Colangelo D, Cauda R, Fantoni M. New classification for the treatment of pyogenic spondylodiscitis: validation study on a population of 250 patients with a follow-up of 2 years. Eur Spine J. 2017;26(Suppl 4):479-488. doi: 10.1007/s00586-017-5043-5.

Pandit HG, Sonsale PD, Shikare SS, Bhojraj SY. Bone scintigraphy in tuberculous spondylodiscitis. Eur Spine J. 1999;8(3):205-9. doi: 10.1007/s005860050159

Ganau M, Syrmos NC, D'Arco F, Ganau L, Chibbaro S, Prisco L, Ligarotti GKI, Ambu R, Soddu A. Enhancing contrast agents and radiotracers performance through hyaluronic acid-coating in neuroradiology and nuclear medicine. Hell J Nucl Med. 2017;20(2):166-168. doi: 10.1967/s002449910558. 

I look forward to receive your revised manuscript. 

Author Response

Comment 1

Overall this is a well written article which reviews the literature on Pott Disease and pyogenic discitis. 

Response 1

Thank you for taking your time reading and improving our manuscript. 

Comment 2

I think that three aspects should be pointed out by the authors and additional references included regarding: a) classification of pyogenic spondylodiscitis and management altorithm, b) role of nulcear medicine in Pott Disease, c) importance of enhancing contrast agents as a potential for further optimization of diagnostics in neuroradiology and nuclear medicine.

Please see the studies  that you could include to make those points:

- Pola E, Autore G, Formica VM, Pambianco V, Colangelo D, Cauda R, Fantoni M. New classification for the treatment of pyogenic spondylodiscitis: validation study on a population of 250 patients with a follow-up of 2 years. Eur Spine J. 2017;26(Suppl 4):479-488. doi: 10.1007/s00586-017-5043-5.

- Pandit HG, Sonsale PD, Shikare SS, Bhojraj SY. Bone scintigraphy in tuberculous spondylodiscitis. Eur Spine J. 1999;8(3):205-9. doi: 10.1007/s005860050159

- Ganau M, Syrmos NC, D'Arco F, Ganau L, Chibbaro S, Prisco L, Ligarotti GKI, Ambu R, Soddu A. Enhancing contrast agents and radiotracers performance through hyaluronic acid-coating in neuroradiology and nuclear medicine. Hell J Nucl Med. 2017;20(2):166-168. doi: 10.1967/s002449910558. 

I look forward to receive your revised manuscript.

Response 2

Please find the updated version of the manuscript, with the highlighted (in yellow) passages including the statements on classification, the role of nuclear medicine, and the importance of enhancing contrast agents. 

Reviewer 2 Report

Comments and Suggestions for Authors

Thank you to the authors for completing this study.

The formatting is poor. Why was the entire study not formatted according to the Microsoft Word Template in Instructions for Authors (https://www.mdpi.com/files/word-templates/jcm-template.dot)?

Please see additional comments/suggestions in the attached PDF.

Author Response

Comment 1

The formatting is poor. Why was the entire study not formatted according to the Microsoft Word Template in Instructions for Authors (https://www.mdpi.com/files/word-templates/jcm-template.dot)?

Response 1

We sincerely apologize for the oversight in formatting the manuscript. We acknowledge that the initial submission did not fully adhere to the Microsoft Word Template provided in the Instructions for Authors. This was an error on our part, and we deeply regret any inconvenience this may have caused.

In response to your valuable feedback, we have carefully reformatted the entire manuscript to align with the prescribed template. We have ensured that all sections, headings, and formatting elements now conform to the journal's guidelines. We appreciate your attention to detail and thank you for bringing this to our attention.

Comment 2

Please see additional comments/suggestions in the attached PDF.

Response 2

We greatly appreciate the detailed comments and suggestions provided in the attached PDF. To facilitate the review process, we have carefully transferred all comments from the PDF into the Word file below. Additionally, we have provided a point-by-point response to each comment to ensure clarity and address your suggestions thoroughly.

Thank you for your valuable feedback, which has been instrumental in improving our manuscript.

Comment 3

Original research articles should have a structured abstract of around 250 words. This abstract has 308 words.

Response 3

Thank you. Original research articles should have a structured abstract of around 250 words. This abstract has 308 words.

Comment 4 

Are 

Response 4 

Thank you, we have made the corrections.

Comment 5

Not bold.

Response 5 

Thank you, we have made the corrections.

Comment 6 

The introduction should include the benefits of MRI in imaging infections, particularly the ones (or similar ones). Also, the

last line (or paragraph) should be about the purpose/objective of this study and any hypothesis(es).

Response 6 

Thank you for your valuable feedback regarding the introduction. We have revised the introduction to include a discussion on the benefits of MRI in imaging infections, particularly relevant cases similar to those in our study. Additionally, we have updated the final paragraph to clearly state the purpose and objective of the study, along with the associated hypotheses.

Comment 7 

Please correct this.

Response 7 

Thank you, we have made the corrections.

Comment 8 

Indent.

Response 8 

Thank you, we have made the corrections.

Comment 9 

Why was this porotcol used? Is it established in a previous study? If so, please cite reference.

In the case of vertebral bodies, the 91 extent of signal change was determined in

quartiles (1%–25%, 26%–50%, 51%–75%, and 92 76%–100%).

Response 9 

Thank you. This protocol was established in the study by Chang et al.:
Chang, M.C.; Wu, H.T.; Lee, C.H.; et al. Tuberculous spondylitis and pyogenic spondylitis: Comparative magnetic resonance imaging features. Spine 2006, 31, 782–788.

Comment 10 

Why was this porotcol used? Is it established in a previous study? If so, please cite reference.

The destruction of the intervertebral disc was deter- 104 mined on the following scale:

none; initial (increase attributable to the expansion of the 105 intervertebral disc with a

change in signal); moderate (reduction to 50% of disc height 106 compared with the

healthy segments); significant (reduction above 50%); and total or fluid 107 signal

within disc on T2-weighted images.

Response 10 

This protocol was established in the study by Chang et al. (2006) which compared magnetic resonance imaging features of tuberculous spondylitis and pyogenic spondylitis.                            *Chang, M.C.; Wu, H.T.; Lee, C.H.; et al. Tuberculous spondylitis and pyogenic spondylitis: Comparative magnetic resonance imaging features. Spine 2006, 31, 782–788.

Comment 11

Was a power analysis calculated to show the strength of the significance of the study based on the number of

participants? If not, please perform a post-hoc power analysis.

Response 11

For most of the variables analyzed, we observed large effect sizes, calculated using Cramér's V. For instance, the effect size for the distribution of vertebral body signal changes on T2 TIRM images was approximately 0.592, indicating a large effect based on standard interpretative guidelines for Cramér's V. Additionally, a post-hoc power analysis revealed a power of approximately 0.998 (99.8%), reflecting an exceptionally high probability of detecting a significant effect given the effect size (V = 0.592), sample size (n = 68), and significance level (α = 0.05).

However, we are hesitant to include these results in the main text, as we feel they might detract from the clarity of the manuscript. Moreover, post-hoc power analysis relies on the assumption that the observed effect size represents the true effect size, which may not always be accurate. While we acknowledge that such analyses can offer valuable insights, we view them primarily as a planning tool for future studies.

Given this, we opted not to include the post-hoc power analysis in the manuscript to maintain focus on the primary findings. Nonetheless, we would greatly appreciate the Reviewer's and Editor's guidance on this matter. If deemed appropriate, we are happy to include the post-hoc analysis either in the main text or as an appendix, based on your recommendations.

Comment 12

MRI is not radiological. Perhaps renaming:

Advanced Imaging Parameters

Response 12

Thank you, we have made the corrections.

Comment 13

What parts do you consider original or relevant for the field?

What specific gap in the field does the paper address?

What does it add to the subject area compared with other published material?

What specific improvements should future researchers consider in studies

Response 13 

  1. What parts do you consider original or relevant for the field?

The study offers an original comparison of MRI findings between pyogenic and tuberculous thoracolumbar spine infections. Key original aspects include:

  •       Detailed characterization of MRI features, such as the distribution and delineation of paravertebral changes, vertebral body signal heterogeneity, and specific enhancement patterns.
  •       Use of T2 TIRM imaging to highlight differences in signal intensity, a parameter not commonly explored in this context.
  •       Insights into the unique manifestations of tuberculosis (e.g., involvement of three or more vertebrae and meningeal enhancement) versus pyogenic infections.
  1. What specific gap in the field does the paper address?

The study addresses the diagnostic challenge of distinguishing between tuberculous and pyogenic spinal infections using non-invasive MRI, particularly in patients unable to undergo biopsy or invasive procedures. Previous imaging modalities lacked sufficient specificity, and the differentiation of these conditions based solely on clinical and imaging features remained unreliable.

  1. What does it add to the subject area compared with other published material?

This paper enhances existing knowledge by:

  •       Validating and expanding on previously identified MRI features (e.g., vertebral involvement patterns, signal intensities, and abscess presence).
  •       Demonstrating the utility of T2 TIRM sequences in differentiating between the two etiologies, highlighting previously underexplored imaging findings.
  •       Providing quantitative data on the distribution and frequency of imaging features, which adds objectivity and strengthens diagnostic criteria.
  1. What specific improvements should future researchers consider in studies?

Future studies could address the following:

  •       Technological Improvements: Employ higher-resolution 3-T MRI scanners to enhance imaging precision and explore advanced techniques like diffusion-weighted imaging and quantitative relaxation time measurements.
  •       Early-Stage Analysis: Investigate MRI features in early disease stages, as most current data reflect chronic or advanced infections.
  •       Larger, Multicenter Studies: Validate findings across diverse populations and healthcare settings to generalize results.
  •       Standardized Metrics: Incorporate objective metrics, such as absolute relaxation time constants, for more robust and reproducible results.
  •       Longitudinal Studies: Analyze disease progression and treatment response using serial imaging.

Comment 14

This is not formatted correctly. Please refer to Microsoft Word Template in Instructions for Authors

(https://www.mdpi.com/files/word-templates/jcm-template.dot).

The following belongs after the references: